# Machine learning based CRISPR gRNA design for therapeutic exon skipping

**Wilson Louie**[1,2], **Max W. Shen**[3], **Zakir Tahiry**[4], **Sophia Zhang**[4], **Daniel Worstell**[4], **Christopher A. Cassa**[4], **Richard I. Sherwood**[4,5]*, **David K. Gifford**[1,2,6]*

**1** Computer Science and Artificial Intelligence Laboratory, Massachusetts Institute of Technology, Cambridge, Massachusetts, United States of America, **2** Department of Electrical Engineering and Computer Science, Massachusetts Institute of Technology, Cambridge, Massachusetts, United States of America, **3** Computational and Systems Biology Program, Massachusetts Institute of Technology, Cambridge, Massachusetts, United States of America, **4** Division of Genetics, Department of Medicine, Brigham and Women's Hospital and Harvard Medical School, Boston, Massachusetts, United States of America, **5** Hubrecht Institute for Developmental Biology and Stem Cell Research, Royal Netherlands Academy of Arts and Sciences (KNAW), Utrecht, The Netherlands, **6** Department of Biological Engineering, Massachusetts Institute of Technology, Cambridge, Massachusetts, United States of America

* rsherwood@rics.bwh.harvard.edu (RIS); gifford@mit.edu (DKG)

**Data Availability Statement:** High-throughput sequencing data have been deposited in the NCBI Sequence Read Archive database under accession code SRP272657

## Abstract

Restoring gene function by the induced skipping of deleterious exons has been shown to be effective for treating genetic disorders. However, many of the clinically successful therapies for exon skipping are transient oligonucleotide-based treatments that require frequent dosing. CRISPR-Cas9 based genome editing that causes exon skipping is a promising therapeutic modality that may offer permanent alleviation of genetic disease. We show that machine learning can select Cas9 guide RNAs that disrupt splice acceptors and cause the skipping of targeted exons. We experimentally measured the exon skipping frequencies of a diverse genome-integrated library of 791 splice sequences targeted by 1,063 guide RNAs in mouse embryonic stem cells. We found that our method, SkipGuide, is able to identify effective guide RNAs with a precision of 0.68 (50% threshold predicted exon skipping frequency) and 0.93 (70% threshold predicted exon skipping frequency). We anticipate that SkipGuide will be useful for selecting guide RNA candidates for evaluation of CRISPR-Cas9-mediated exon skipping therapy.

## Author summary

One form of genetic therapy is exon skipping, where a cell is forced to exclude problematic exons from a mutant transcript such that the resultant protein is functional. Recent studies show that CRISPR technology can induce therapeutic exon skipping. By using a specific guide RNA, targeted disruption of an exon's splice acceptor sequence can be performed, which can result in its skipping. However, an exon may have many candidate guide RNAs that target its splice acceptor, and not all guide RNAs will lead to a sufficient level of exon skipping. A predictive method that can identify a guide RNA that will cause an exon to be skipped would be useful for guiding therapeutic development efforts. We

(https://www.ncbi.nlm.nih.gov/sra/?term=SRP272657). An open source implementation of SkipGuide is available at https://github.com/gifford-lab/skipguide. Code for data processing, analysis, and production of the results, figures, and tables is available at https://github.com/gifford-lab/skipguide-analysis.

**Funding:** The study was supported by the National Institutes of Health: 1R01HG008363, 1R01HG008754 (to D.K.G.). The funders had no role in study design, data collection and analysis, decision to publish, or preparation of the manuscript.

**Competing interests:** The authors have declared that no competing interests exist.

present SkipGuide, a machine learning method for predicting the exon skipping level caused by a guide RNA that targets its splice acceptor region. To develop and evaluate SkipGuide, we experimentally measured the skipping levels of a diverse set of exons targeted by multiple guide RNAs in a mouse cell line. We demonstrate that SkipGuide can accurately identify the guide RNAs that lead to high levels of exon skipping.

This is a *PLOS Computational Biology* Methods paper.

## Introduction

Exon skipping therapies have emerged as a powerful method for treating genetic disorders by restoring gene function. These therapies work by forcing the RNA splicing machinery to bypass exons that contain deleterious point or frameshift mutations. The first clinical success in exon skipping disease treatment uses antisense oligonucleotide strategies to alter RNA splicing, and has recently received FDA approval for treatment of Duchenne muscular dystrophy (DMD) [1, 2]. Several other disease exon skipping via antisense oligonucleotide treatments have shown pre-clinical and clinical promise, and are in development for diseases including cystic fibrosis [3], atherosclerosis [4], cardiomyopathy [4–7], and Pompe Disease [8]. However, oligonucleotide therapies are not only expensive, but also require lifelong dosing due to their transient nature [9].

CRISPR (Clustered regularly interspaced short palindromic repeats) editing of genomic regulatory elements that control exon splicing is a promising means of permanently alleviating genetic diseases. CRISPR-Cas9 genome editing can be used to ablate the functioning of exon splicing sequences, causing the RNA splicing machinery to skip a deleterious exon. Following Cas9-induced DNA double-stranded break, non-homologous end joining and microhomology-mediated end joining repair pathways often introduce variable indel mutations at the cut site [10, 11]. By using an appropriate guide RNA (gRNA) to direct Cas9 to induce a double-stranded break at the splice donor or acceptor sequence of a mutant exon, end-joining repair can disrupt the genomic sequence of these splice sites sufficiently to cause skipping of the mutant exon. This strategy of template-free Cas9 induced exon skipping has been shown to provide efficient and permanent phenotypic alleviation in cellular and animal models of DMD [12–14]. This strategy is also more therapeutically practical than an approach of CRISPR-Cas9-mediated mutation reversion using template-directed homology-directed repair, because it is limited by low efficiency (particularly in postmitotic adult tissues), undesired by-products, and the need to deliver a DNA repair template [15–17].

An exon can have many candidate gRNAs for exon skipping, and the existence of an effective gRNA for a particular exon is not guaranteed or known *a priori*. A predictive method that can identify effective gRNAs for a target exon would thus be useful for guiding therapeutic development efforts.

In this study, we describe a machine learning pipeline, SkipGuide, that predicts the skipping level of an exon given a *Streptococcus pyogenes* Cas9 (*Sp*Cas9) gRNA that targets its splice acceptor site. In conformity with notations of previous literature [18–21], we quantify exon skipping as the percent spliced-in (PSI, $\Psi$), defined as the fraction of transcripts that contains the exon. Thus, the frequency of exon skipping is $1 - \Psi$. SkipGuide employs inDelphi [22] to

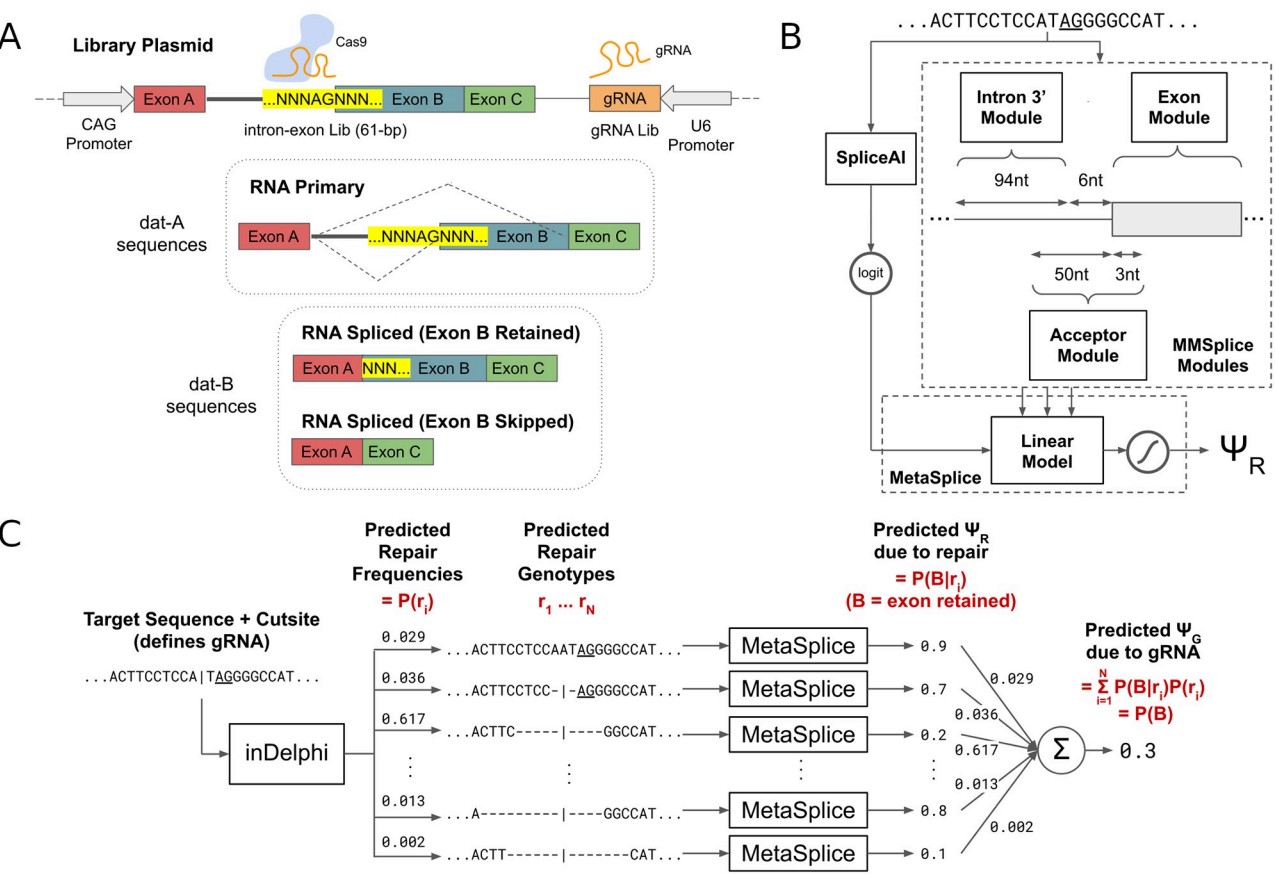

**Fig 1. Genome-integrated reporter system, and machine learning pipeline for predicting exon skipping levels from *Sp*Cas9 gRNAs that target splice acceptors.** (A) A simplified illustration of the genome-integrating high-throughput CRISPR-Cas9 splicing reporter system. (B) MetaSplice combines predictions from SpliceAI, and MMSplice's intronic, acceptor site, and exonic modules using a linear model tuned on experimental data from (A). (C) The full machine learning pipeline, SkipGuide, predicts $\Psi_G$ for a given *Sp*Cas9 gRNA, by chaining inDelphi and MetaSplice. Probabilistic interpretations are shown in red.

predict DNA repair genotypes and frequencies caused by Cas9 in combination with a given *Sp*Cas9 gRNA, and a linear model we call MetaSplice, that combines predictions from SpliceAI [23], and the intronic, acceptor site, and exonic modules of the Modular Modeling of Splicing framework (MMSplice) [24], to predict $\Psi$ for each repair genotype ($\Psi_R$, Fig 1B). The overall predicted $\Psi$ for a given gRNA ($\Psi_G$) is then taken to be the mean predicted $\Psi_R$ over all its predicted repair genotypes, weighted by the predicted frequency of each repair genotype (Fig 1C). inDelphi has demonstrated high accuracy at predicting *Sp*Cas9 genotypic outcomes [22]. SpliceAI is a splice junction prediction model shown to outperform other models such as GeneSplicer [25] and NNSplice [26]. MMSplice is a model of exon skipping ranked first at the recent fifth Critical Assessment of Genome Interpretation group [27, 28], shown to outperform state-of-the-art models such as COSSMO [20], HAL [19], SPANR [18], and MaxEntScan [29].

To evaluate SkipGuide, we constructed a diverse genome-integrated library of splice acceptor sequences targeted by multiple gRNAs in mouse embryonic stem cells (mESCs), and experimentally measured $\Psi_G$ resulting from each gRNA (Fig 1A). We found that SkipGuide is able to identify gRNAs that lead to low $\Psi_G$ with high precision. Thus, we expect our method to facilitate studies in CRISPR-Cas9-mediated therapeutic exon skipping, by enabling *a priori* selection of likely effective gRNAs.

## Results

### Genome-integrated high-throughput assay for CRISPR-Cas9-mediated exon skipping

To observe Cas9-mediated end-joining repair products and their eventual $\Psi_R$ across a wide variety of intron-exon junctions, we designed a genome-integrated gRNA and intron-exon target library (lib-SA), in which each 61 base pair target site is accompanied by a corresponding $Sp$Cas9 gRNA on the same DNA molecule (Fig 1A and S1 Fig and S1 Text). To explore effects representative of the human genome, we designed 1,927 target sites derived from human intron-exon junctions, each targeted by at least one gRNA (4,000 total, see Methods). Each construct includes three Exons (A, B, and C) that are part of a single mRNA transcript, and the assay is designed to produce sequencing reads that indicate the observed frequency of Exon B skipping. Each of the 1,927 target sites is a splice acceptor for Exon B, and a constant alternative splice acceptor is at the Exon B and Exon C boundary (S1 Text). Depending on the severity of splice acceptor disruption from Cas9-mediated repair at the target site, differential splicing of the resultant primary RNA transcripts may favor the skipping of Exon B (Fig 1A). Thus, the $\Psi_R$ of a particular repair outcome refers to the fraction of their transcripts that contain Exon B.

We stably integrated lib-SA into the genomes of mESCs, treated the cells with Cas9, and performed paired-end high throughput DNA sequencing on the gRNA sequence of lib-SA, the primary RNA transcripts, and the spliced RNA transcripts. We also sequenced control cells prior to Cas9 treatment. We processed the resulting sequencing data (see Methods, and S9 Fig) to identify and quantify the genomic repair outcomes associated with each gRNA, and to identify the presence or absence of Exon B in all transcripts associated with each repair outcome.

The resulting dataset that identifies the genotypic outcomes of splice acceptor repair we call 'dat-A', and the dataset that identifies the skipping frequency we call 'dat-B'. Dataset dat-A is constructed from 1,998,925 primary transcript sequencing reads. It represents 1,695 gRNAs that target 1,549 lib-SA targets, and consists of a mean of 45 unique repair genotypes per associated gRNA (S2 Table). Since primary transcript sequences reflect the genomic repair genotype of lib-SA targets, we estimated the frequency of each unique repair genotype from a mean of 1,180 primary transcript sequencing reads per associated gRNA. The dataset dat-B is constructed from 520,196 spliced transcript sequencing reads. It represents 1,063 gRNAs that target 791 lib-SA targets, and consists of a mean of 246 spliced transcript sequencing reads per associated repair outcome, for 2,113 unique repair outcomes (S3 Table). From this dataset, $\Psi_R$ for each of the 2,113 repair outcomes was computed, and by aggregating the spliced transcript sequencing reads by associated gRNA, the $\Psi_G$ associated with each of the 1,063 gRNA was computed. A similar dataset, dat-B WT, was constructed from 3,443,915 spliced transcript sequencing reads of control cells before Cas9 treatment (S4 Table). The wild type Exon B skipping level (WT $\Psi$) for 1,697 lib-SA targets were derived from this dataset.

We report the performance of inDelphi in predicting the observed repair outcomes in dat-A, the performance of MetaSplice in predicting $\Psi_R$ derived from dat-B, the comparison between $\Psi_G$ and WT $\Psi$, and the performance of SkipGuide in predicting the observed $\Psi_G$ from the associated gRNAs.

### inDelphi accurately predicts genotypes and frequencies of experimentally observed Cas9-mediated end-joining repair products

From dat-A, we found that end-joining repair of Cas9-mediated DNA cuts of our lib-SA in mESCs primarily led to deletions (83.3% of all products) and single base pair (1-bp) insertions

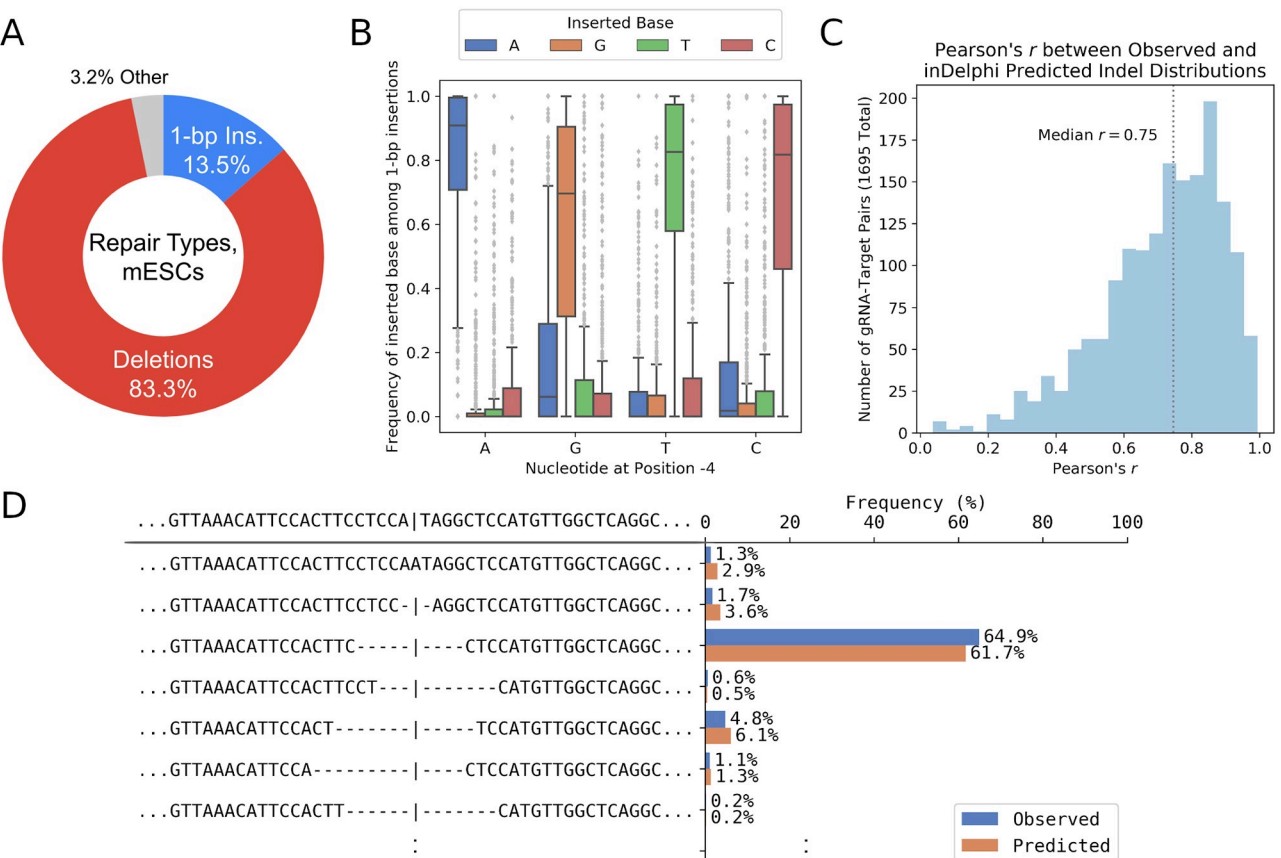

**Fig 2. inDelphi accurately predicts editing outcomes.** (A) The categories of editing products at 1,549 lib-SA target sites in mESCs. "Other" denotes products such as multiple base pair insertions and combination indels. (B) 1-bp insertion frequencies among 1,549 lib-SA target sites, compared to the −4 nucleotide from their NGG PAM. (C) The distribution of Pearson's *r* between observed and inDelphi predicted repair genotype frequencies for 1,695 gRNA lib-SA target pairs. (D) An example distribution of observed and inDelphi predicted repair genotype frequencies (*r* = 0.98) for a particular gRNA lib-SA target pair (S1 Table, Identifier number 5026).

(13.5% of all products) that overlap the cutsite (Fig 2A). The remaining 3.2% of repair outcomes consist of multiple base pair insertions at the cutsite, combinations of insertions and deletions that overlap the cutsite, and insertions and/or deletions (indels) that do not overlap the cutsite.

To ensure that a meaningful evaluation of inDelphi's accuracy and generalizability is possible, we first confirmed that the sequences used to train inDelphi [22] do not overlap, and are not homologous to, the lib-SA target sequences that make up dat-A. Through a pairwise local sequence alignment procedure (see Methods), we found that the distribution of the best-aligned sequence identities between lib-SA sequences in dat-A and inDelphi's training set is right-tailed (S2 Fig). Except for one lib-SA sequence in dat-A (out of 1,549) that shares 67% sequence identity with a sequence from inDelphi's training set, all lib-SA sequences in dat-A share less than 37% sequence identity with inDelphi's training set, with a median and mean (best-aligned) sequence identity of 24% and 23% respectively.

We then compared inDelphi's indel frequency predictions and the observed genotypes in dat-A, and found that there is a high correlation between the two (median *r* = 0.75) across the 1,695 gRNAs represented (Fig 2C and 2D). For each of the 1,695 gRNAs, we considered only repair genotypes that resulted from either 1 to 60-bp deletions spanning the cutsite, or 1-bp

insertions at the cutsite. These are the repair types that inDelphi models [22], and they encompass all detectable repairs on our 61-bp lib-SA target sequences.

Since inDelphi internally uses separate models for 1-bp insertion predictions and deletion predictions, we also assessed the accuracy of inDelphi's insertion and deletion predictions separately. For predicting 1-bp insertions, inDelphi employs a nearest neighbors model that assumes genotypes of 1-bp insertions are local sequence context dependent, and are predominantly duplications of the −4 nucleotide from the NGG *Sp*Cas9 protospacer-adjacent motif (PAM) [22]. We found that this inductive bias generalizes to our measured 1-bp insertion frequencies among 1,549 lib-SA target sites (Fig 2B), and inDelphi's 1-bp insertion predictions accurately reflects the observed 1-bp insertion in dat-A (median $r$ = 0.95, S2 Fig). We also found that there is a high correlation between inDelphi's deletion predictions and observed deletions in dat-A (median $r$ = 0.75, S2 Fig).

## MetaSplice predictions of $\Psi_R$ correlate with those of experimental observations

SkipGuide requires the ability to predict $\Psi_R$ for an individual repair product based upon the sequence of its splice acceptor region. We evaluated MaxEntScan [29], SpliceAI [23], MMSplice [24], and combinations of SpliceAI and MMSplice for this task using the dat-B derived $\Psi_R$ (Fig 3 and S3 Fig). SpliceAI and MMSplice have reported superior accuracy over existing splice junction scoring models [23, 24], and MaxEntScan was considered as a baseline reference.

MaxEntScan scores a 23-nt region surrounding an acceptor site, but does not predict $\Psi_R$, so for MaxEntScan we computed Pearson's $r$ between its score and the observed $\Psi_R$ values. For all other methods we computed Pearson's $r$ between the predicted and observed $\Psi_R$ as well as mean absolute error (*MAE*), mean squared error (*MSE*), and root-mean-square error (*RMSE*). We found that the MaxEntScan scores on the 2,113 repair genotypes in dat-B yield the lowest correlation with the actual observed $\Psi_R$ values compared to the other methods ($r$ = 0.09, Fig 3A and Table 1).

SpliceAI predicts the probability a position is used as a splice acceptor for every position of a given sequence. We used SpliceAI to predict probabilities at every position of each repair genotype in dat-B, and took the maximum probability over all the positions as the predicted $\Psi_R$ for each sequence. We found that the SpliceAI predictions evaluated against the actual $\Psi_R$ values observed in dat-B achieved $r$ = 0.50, *MAE* = 0.29, *MSE* = 0.13, and *RMSE* = 0.36 (Fig 3B and Table 1).

MMSplice consists of separate prediction modules for scoring overlapping splicing-relevant sequence regions. We first evaluated MMSplice's 3' intronic module, acceptor site module, and exonic module separately on dat-B, and found that they yield $r$ = 0.45, $r$ = 0.20, and $r$ = 0.18 respectively (S3 Fig). We then evaluated the performance of a linear model that combines the predictions from these three modules by 10-fold cross validation on dat-B derived $\Psi_R$ (Fig 3 and Methods). We call this linear model Weighted MMSplice (wMMSplice). To construct the folds, we first grouped the 2,113 repair outcomes in dat-B into the 1,063 associated gRNAs, and then randomly partitioned them into 10 folds (S6 Fig). This ensures that no two folds contain repair outcomes from the same associated gRNA. By using each fold as a validation set, and the other 9 folds as training data to fit wMMSplice's linear model weights (via ridge regression [30], with L2 regularization strength of 0.35, see Methods), we produced cross validated predictions of $\Psi_R$ for all 2,113 repair outcomes. When compared with the actual values of $\Psi_R$, we found that wMMSplice predictions achieved mean $r$ = 0.55, mean *MAE* = 0.18, mean *MSE* = 0.06, and mean *RMSE* = 0.25, over the 10 folds (Fig 3C and Table 1). We also

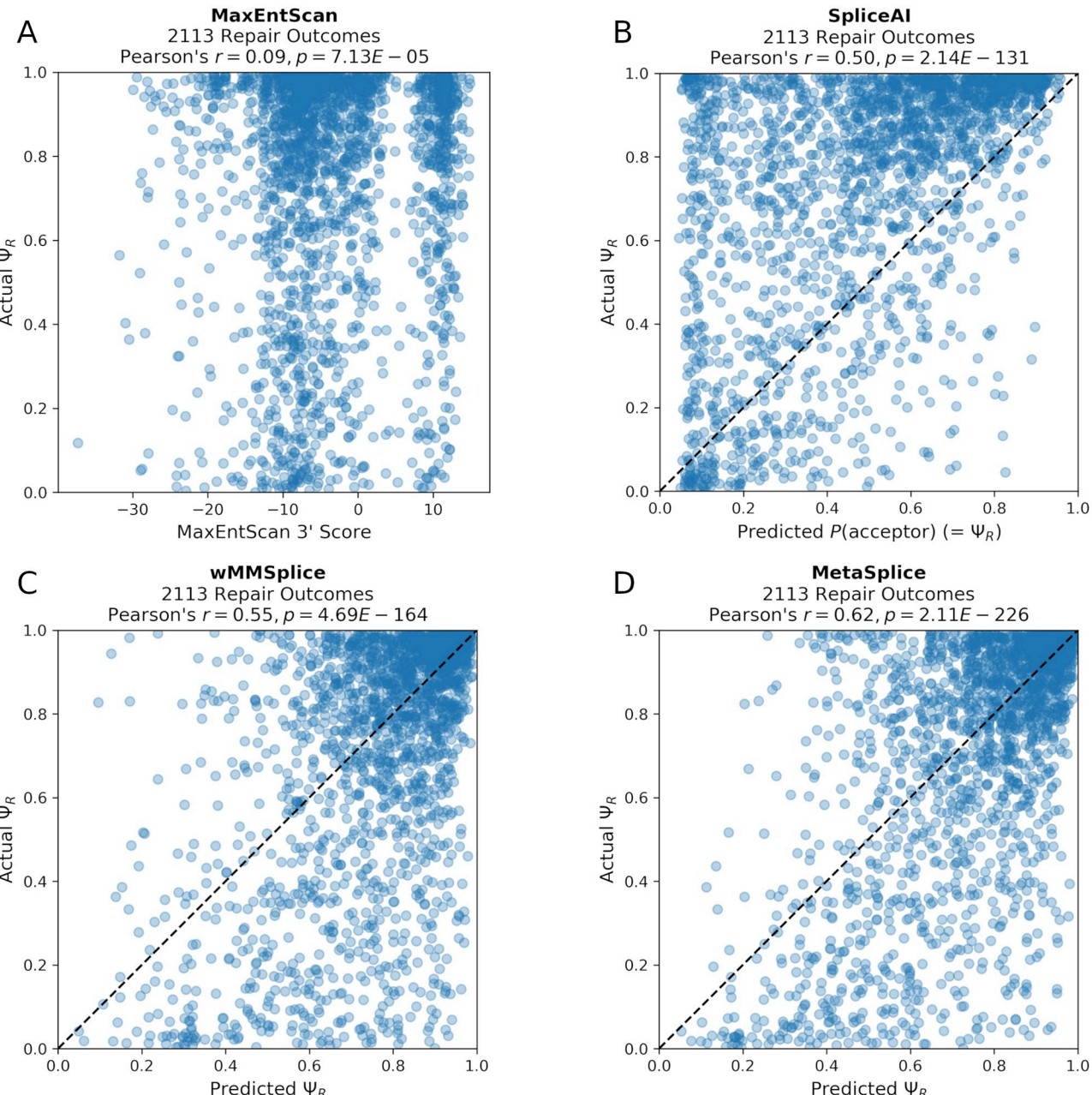

**Fig 3. Actual $\Psi_R$ vs. MaxEntScan scores, SpliceAI predictions, wMMSplice cross validated predictions, and MetaSplice cross validated predictions of $\Psi_R$ for 2,113 repair outcomes.** Each plot shows the actual $\Psi_R$ vs. scores and predictions from various methods on 2,113 different repair outcomes (associated with 1,063 gRNAs). (A) Plot of the actual $\Psi_R$ vs. MaxEntScan scores ($r = 0.09$). (B) Plot of the actual $\Psi_R$ vs. SpliceAI predictions ($r = 0.50$). (C) Plot of the 10-fold cross validated predictions of $\Psi_R$ from wMMSplice (mean $r = 0.55$). (D) Plot of the 10-fold cross validated predictions of $\Psi_R$ from MetaSplice (mean $r = 0.62$). The same 10 folds were used for wMMSplice and MetaSplice.

repeated the 10-fold cross validation 1,000 times, each with a different random split of the folds, and found the same level of performance (S10 Fig).

In a similar fashion as wMMSplice, we evaluated the performance of a linear model that combines the predictions from the three MMSplice modules and those from SpliceAI by 10-fold cross validation (see Methods). We call this linear model MetaSplice. The same data

**Table 1. Performance of $\Psi_R$ predictions by various methods.**

| | MaxEntScan | SpliceAI | wMMSplice | | MetaSplice | |
| --- | --- | --- | --- | --- | --- | --- |
| | | | Mean | SD | Mean | SD |
| Pearson's $r$ | 0.09 | 0.50 | 0.55 | 0.06 | **0.62** | 0.05 |
| MAE | - | 0.29 | 0.18 | 0.01 | **0.16** | 0.02 |
| MSE | - | 0.13 | 0.06 | 0.01 | **0.05** | 0.01 |
| RMSE | - | 0.36 | 0.25 | 0.02 | **0.23** | 0.02 |

Evaluation of models predicting $\Psi_R$ on dat-B. MAE denotes mean absolute error, MSE denotes mean squared error, RMSE denotes root-mean-square error, and SD denotes standard deviation. MaxEntScan scores sequences, but does not predict $\Psi_R$, so only the Pearson's $r$ metric is available. Out of the methods shown, MetaSplice's 10-fold cross validated predictions of $\Psi_R$ yield the highest correlation with, and the lowest error against, the observed $\Psi_R$ in dat-B.

folds and ridge regression parameters used for training and evaluating wMMSplice were used on MetaSplice to produce cross validated predictions of $\Psi_R$ for all 2,113 repair outcomes (see Methods). We found that MetaSplice predictions achieved the highest correlation with, and the lowest error against, the observed $\Psi_R$ in dat-B compared to the other methods: mean $r = 0.62$, mean $MAE = 0.16$, mean $MSE = 0.05$, and mean $RMSE = 0.23$, over the 10 folds (Table 1 and Fig 3D and S10 Fig). We also evaluated MetaSplice on dat-B WT, by first fitting MetaSplice on all the dat-B derived data, and found that the predictions achieved $r = 0.58$ when compared with the actual values of WT $\Psi$ (S4(F) Fig).

## SkipGuide identifies gRNAs that cause effective exon skipping with high precision

We hypothesized that Cas9-mediated genotypic alteration of lib-SA splice acceptors would generally result in lower $\Psi_G$ than wild-type (WT) lib-SA splice acceptors. To investigate this, we considered the set of gRNAs associated with transcripts observed in Cas9 treated cells (S3 Table), and the set of gRNAs associated with transcripts observed in WT cells prior to Cas9 treatment (S4 Table). We identified 735 gRNAs common to both sets with sufficient support for us to compute their $\Psi$ values (see Methods). We found that after Cas9 treatment, all 735 gRNAs resulted in similar or lower $\Psi_G$ compared to their WT counterparts (treated mean $\Psi_G = 0.68$, WT mean $\Psi = 0.91$, see Fig 4A (colorbar) and S5 Fig). Aside from 5 out of 735 with $\Psi$ between 0.45 and 0.5, all of the WT lib-SA targets exhibited $\Psi$ greater than 0.5.

We then evaluated SkipGuide's performance in predicting $\Psi_G$ given the wild type sequence of a gene and the sequence of a gRNA that targets a contained splice acceptor. For each of the the 1,063 gRNAs from dat-B, we predicted the splice acceptor genotypes that would result using inDelphi, the $\Psi_R$ for each genotype using MetaSplice, and the $\Psi_G$ (Fig 1C). We obtained these predictions using the same 10 fold cross validation strategy used to evaluate MetaSplice: $\Psi_G$ predictions for each gRNA in one fold are obtained by using the other 9 folds to fit MetaSplice within SkipGuide, and we repeated this for all 10 folds to obtain predictions for all 1,063 gRNAs (S6 Fig). When compared with the actual values of $\Psi_G$, we found that SkipGuide predictions achieved overall $r = 0.67$, mean $r = 0.67$, mean $MAE = 0.17$, mean $MSE = 0.05$, and mean $RMSE = 0.21$, over the 10 folds (Fig 4A and S7 and S10 Figs).

We found that SkipGuide predictions are more reliable for lower predicted values of $\Psi_G$. To quantify this, we binarized the observed $\Psi_G$ values with a threshold of 0.5. We consider gRNAs with observed $\Psi_G \leq 0.5$ as 'effective' at causing exon skipping, and 'ineffective' otherwise. If an analogous threshold $\tau$ is chosen for SkipGuide's predicted $\Psi_G$, then Fig 4A can be subdivided into quadrants representing regions of true positives (TP), false positives (FP), true negatives (TN), and false negatives (FN). For varying values of $\tau$, we computed the precision of

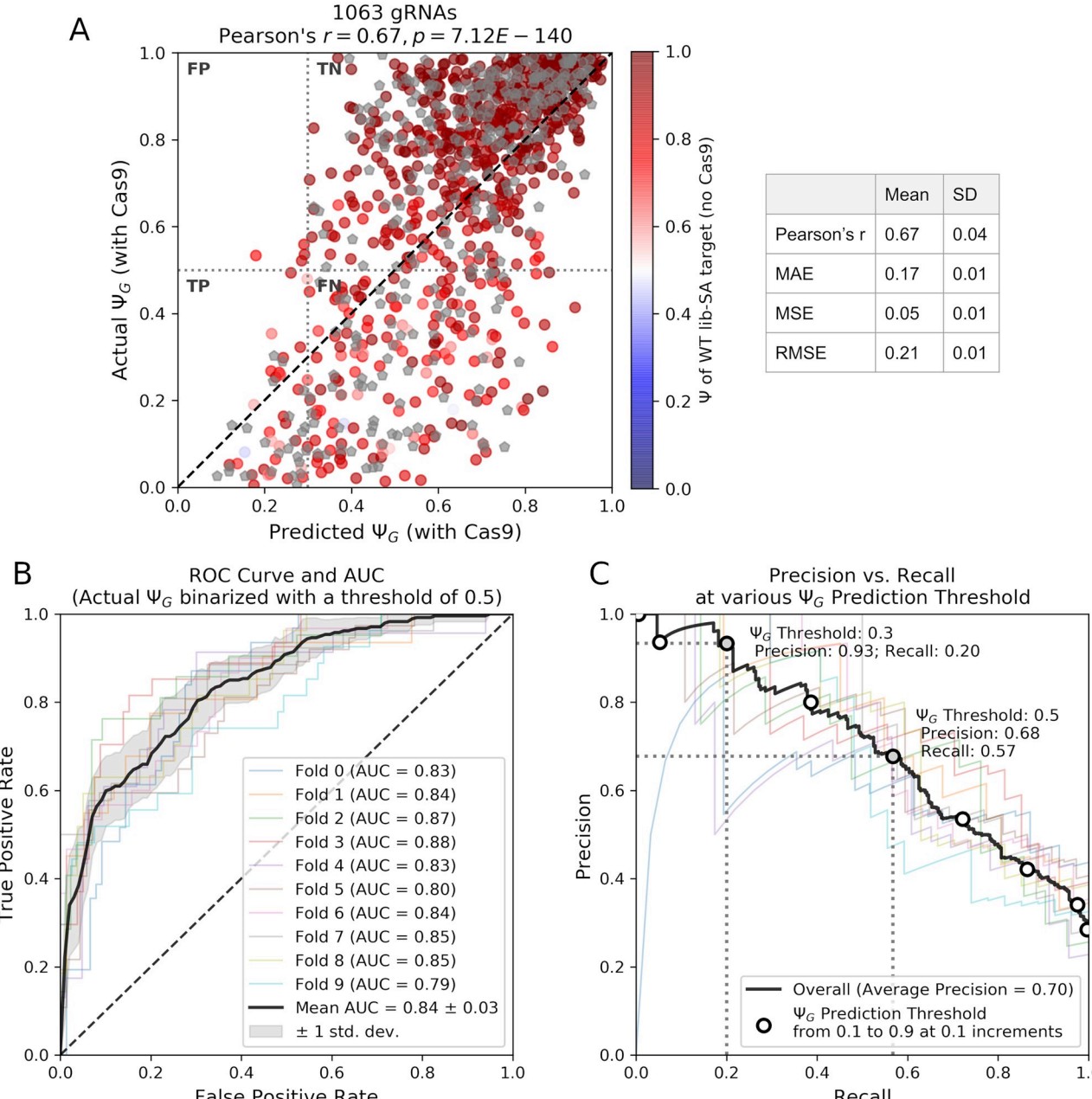

**Fig 4. Performance of SkipGuide in identifying effective *Sp*Cas9 gRNAs.** (A) Plot of the actual $\Psi_G$ vs. the 10-fold cross validated predictions of $\Psi_G$ from SkipGuide on 1,063 gRNAs ($r = 0.67$). Various regression metrics averaged over the 10 folds are shown (MAE denotes mean absolute error, MSE denotes mean squared error, RMSE denotes root-mean-square error, and SD denotes standard deviation). Each point is either colored by the corresponding WT $\Psi$ before Cas9 treatment (735 total), or is labeled with a gray pentagon if the corresponding WT $\Psi$ is unknown. Regions of true positives (TP), false positives (FP), true negatives (TN), and false negatives (FN) are shown when the actual $\Psi_G$ is binarized with a threshold of 0.5, and the predicted $\Psi_G$ is binarized with a threshold of 0.3. (B) Receiver operating characteristic (ROC) curves across the 10 folds, using a 0.5 binary threshold for the actual $\Psi_G$ (mean AUC = 0.84). (C) A plot of precision vs recall (of both overall and each of the 10 folds, assuming the threshold for the actual $\Psi_G$ is 0.5) at various choices of predicted $\Psi_G$ binarization threshold. Choosing a prediction threshold of 0.5 gives a precision of 0.68, and choosing a lower threshold of 0.3 gives a higher precision of 0.93.

the predictions. We found that if treated as a binary classifier, SkipGuide achieved a mean AUC of 0.84 over 10-fold cross validation, and identified effective gRNAs with a precision of 0.68 if $\tau = 0.5$ is used (Fig 4B and 4C). Lower choices of $\tau$ corresponding to a higher skipping frequency yielded higher precision. SkipGuide's high precision at low values of $\tau$ makes it

useful in practice, as one can be more confident that a gRNA predicted to be effective in causing high exon skipping is not a false positive.

Finally, we performed perturbation studies to explore the contributions of inDelphi and MetaSplice to SkipGuide's overall performance (S8 Fig). When we replaced inDelphi's repair frequencies predictions with uniform frequencies, such that SkipGuide's final step is replaced with an unweighted average of MetaSplice predictions, we found that SkipGuide produced predictions with a lower correlation with the actual values of $\Psi_G$ ($r = 0.57$). When we replaced MetaSplice's predictions with random values between 0 and 1, SkipGuide produced predictions with no correlation with the actual values of $\Psi_G$ ($r = 0.03$, $p = 0.30$), with statistically insignificant $p$-values under the null model of $r = 0$. Similarly, when we replaced both inDelphi and MetaSplice with the uniform and random predictors respectively, SkipGuide produced predictions with no correlation with the actual values of $\Psi_G$ ($r = 0.03$, $p = 0.29$). These experiments suggest that both inDelphi and MetaSplice are necessary components for SkipGuide to achieve the best performance.

## Discussion

We find that SkipGuide predicts with high precision the expected skipping frequency of an exon given the sequence of a *Sp*Cas9 gRNA that targets its splice acceptor. To develop and evaluate SkipGuide, we experimentally measured the exon skipping frequencies caused by 1,063 diverse gRNAs that target 791 splice acceptor sequences in mESCs. SkipGuide has the highest precision for high skipping frequencies. Lower skipping frequency thresholds can be used to trade off lower precision with larger sets of candidate gRNAs. The number of diseases potentially treatable by exon skipping therapy is unknown. In this study (see Methods), we applied a narrow set of constraints on the HGMD [31], Ensembl [32], and Pfam [33] databases to identify 1,927 exons in 923 unique genes with known pathogenic indels potentially amenable to CRISPR-Cas9 (acceptor site ablation) mediated exon skipping treatment. These sequences make up the lib-SA library provided in this work, and may be of interest to the community for further evaluation.

While inDelphi [22] is sufficiently accurate in our experimental context, we found MetaSplice to be the limiting factor in SkipGuide's performance. MetaSplice is a meta-model that combines predictions from state-of-the-art models MMSplice [24] and SpliceAI [23]. MetaSplice's relatively low recall in our evaluations reflects the field's incomplete understanding of the biology of splicing and current limited predictive capabilities. However in practice, the region of interest for exon skipping is for small $\Psi$, where MetaSplice, and by extension, SkipGuide, shows high precision. The level of precision we provide provides practically important information for gRNA selection for exon skipping, since all gRNAs selected using our model can be experimentally assayed for this property.

Since acceptor site alteration allows reliable control over which exon is skipped, we focused on the effects of CRISPR-Cas9 mediated acceptor site disruption on exon skipping in this study. SkipGuide is evaluated on gRNAs that lead to DNA double-stranded breaks in a small window (6-bp) surrounding and including the AG splice site acceptor motif. End-joining repair at cut sites outside of this region [34–37], and ablations of an exon's splice donor sequence [37, 38], have been shown to also lead to exon skipping. Hence, modelling these cases can be future extensions to SkipGuide.

The cellular interpretation of splice acceptor sequences and the resulting exon skipping frequencies is also known to be tissue-specific [39, 40], but SkipGuide in this study is only evaluated in the context of mESCs with *Sp*Cas9. inDelphi models only a small number of cell types [22], MMSplice [24] and SpliceAI [23] are not tissue specific, and MetaSplice is tuned using

our data from mESCs. However, SkipGuide is a modular pipeline, so its components can be replaced with more accurate or suitable models in the future. Our experimental design can also be applied in different cell types to provide data for training cell type specific versions of MetaSplice.

We validated the use of gRNAs with *Sp*Cas9 to ablate genome-integrated sequences (dat-A) that lead to changes in exon skipping levels (dat-B) in their native cellular context, and demonstrated SkipGuide prediction generalizability through cross validation. Though we have not validated SkipGuide predictions on native endogenous exons other than our genome integrated constructs, it has been shown that the results from a similar high-throughput assay used in the inDelphi study match edits at endogenous loci [22]. Therefore, we expect that CRISPR-Cas9 mediated acceptor sequence disruption will work endogenously.

A clinically successful gRNA would induce high efficiency of exon skipping with minimal to no off-target effects, but we note that SkipGuide does not consider gRNA off-target and on-target editing efficiency. One way to address this in practice is to use other predictive tools such as Azimuth [41] and Elevation [42] to produce a set of candidate gRNAs with predicted minimal off-target and maximal on-target activities respectively prior to using SkipGuide. Off-target and on-target editing for candidate therapeutic gRNAs from SkipGuide will need to be experimentally evaluated.

The prediction of CRISPR-Cas9 induced exon skipping not only allows *a priori* selection of promising gRNAs for skipping an exon of interest, but also enables identification of candidate exons amenable to this method of induced skipping for further study. Thus, we anticipate SkipGuide will be useful for guiding future research in CRISPR-Cas9 mediated exon skipping therapy.

## Methods

### High-throughput CRISPR-Cas9 splicing reporter system

**Library design: Identifying human intron-exon sequences for lib-SA.** To model exon splicing changes representative of the human genome, we curated human intron-exon junction sequences for the lib-SA target library. This was done by selecting exons that contain at least one pathogenic indel in the Human Gene Mutation Database (HGMD, professional release 2018.1) [31] with the following properties: a basal frameshift rate of 66% or more, which are likely to disrupt normal protein function; a length evenly divisible by 3, to preserve the reading frame when skipped; are not constitutive, as defined by less than 100% presence in Ensembl (release 92) transcripts [32]; and do not contain an annotated protein domain in Pfam (release 31) [33]. This resulted in 6,805 intron-exon sequences. Those without a suitable gRNA for effective targeting (described in the next section) were filtered out, and resulted in the final lib-SA target library of 1,927 human intron-exon sequences (S1 Table).

**Library design: Selecting gRNAs with effective targeting of lib-SA targets.** For each of the candidate intron-exon sequences, candidate gRNAs were identified by considering gRNAs with suitable CRISPR Cas9 cut sites, as defined by the existence of an NGG PAM sequence in a 6-bp window surrounding and including the AG splice site acceptor motif (or the intron-exon boundary if the splice site was not a canonical AG). Azimuth [41] and BOTM (see S2 Text) were used to retain only those gRNAs predicted to have high on-target editing efficiency (Azimuth score above 0.2 and BOTM score above 0.65).

This candidate set of gRNAs was then heuristically reduced using inDelphi [22] and Max-EntScan [29]. inDelphi, initialized to use its mESC models, was used to predict the frequency distribution of repair genotypes for each gRNA. As per inDelphi's default settings, only predictions for 1-bp insertions and between 1-bp and 60-bp microhomology deletions were

considered. For each predicted repair genotype, MaxEntScan's score3ss module was used to estimate the splice site acceptor motif strength. A genotype was classified as motif disrupting if its MaxEntScan score is less than 0.9, and no effect otherwise, as per previous studies on this classification ruleset [43]. The total frequency of all motif-disrupted repair genotypes weighted by the inDelphi predicted repair genotype frequency was taken to be the predicted frequency of splice site disruption. The top 4,000 gRNAs, and their 1,927 associated intron-exon sequences with high predicted frequencies were selected for our library (S1 Table).

**Library cloning, cell culture, and deep sequencing.** lib-SA consists of a CAGGS (CAG) promoter [44] driving a fixed Exon A with a strong splice-donor site, an intron, a variable Exon B, a fixed Exon C, and a polyA sequence (S1 Fig). Our lib-SA of 61-bp intron-exon sequences span the intron-Exon B boundary with 37-nt of intronic sequence and 24-nt of exonic sequence. A highly diverse 15-bp barcode is embedded in Exon C. A U6 promoter on the same DNA molecule drives the corresponding SpCas9 gRNA spacer. The nucleotide resolution description of this construct is provided in S1 Text.

We used a similar library cloning and cell culture procedure to the inDelphi study [22]. lib-SA was constructed through a multistep process, cloned into a plasmid backbone allowing Tol2 transposon-based integration into the genome (S3 Text), and integrated into the genomes of mESCs using Lipofectamine 3000 transfection along with equal molar Tol2 transposase followed by one week of Hygromycin selection to ensure genomic integration.

One week after library integration, the mESCs were transfected with p2T CAG Cas9 BlastR (Addgene 107190) and Tol2 transposase using Lipofectamine 3000 followed by one week of Blasticidin selection to maximize Cas9 activity. After one week, genomic DNA (gDNA) (Pure-link Genomic DNA mini kit) and RNA (Qiagen RNEasy Maxi kit) were extracted from separate aliquots of each replicate culture. gDNA and RNA from cells prior to Cas9 treatment were also extracted as control. Samples were prepared for Illumina Nextseq using PCR-based methods (S3 Text), and paired-end high-throughput DNA sequencing (Illumina Nextseq 2 x 75-nt kit) was then performed on the gDNA and RNA (S1 Fig primer locations, and S3 Text). Technical replicate sequencing was performed on samples from post Cas9 exposed cells.

## Processing of sequencing reads

Read sets from three different types of molecules were used: gDNA (Library Plasmid reads), RNA Un-Spliced, and RNA Spliced as shown in S1 Fig. The gDNA sequencing reads captured barcodes and associated gRNA sequences. The RNA Un-Spliced reads captured barcodes and their associated splice acceptors, and were observed both pre and post Cas9 exposure. The RNA Spliced reads captured barcodes and the corresponding presence or absence of Exon B, and were observed both pre and post Cas9 exposure.

More specifically, there were: 1 pre Cas9 gDNA read set, 2 post Cas9 gDNA read sets from technical replicates, 1 pre Cas9 RNA Un-Spliced read set, 2 post Cas9 RNA Un-Spliced read sets from technical replicates, 1 pre Cas9 RNA Spliced read set, and 2 post Cas9 RNA Spliced read sets from technical replicates.

**Sequence and barcode extraction.** Sequences (gRNA sequence in gDNA, the target sequence in RNA Un-Spliced, the Exon B portion of the target sequence, or lack thereof, in RNA Spliced) and the associated barcode sequences were extracted from sequencing reads using regular expression pattern matching with at most 5 substitutions permitted (S4 Text). Python's `regex` package was used for this purpose. Refer to S4 Text for the specific regular expression patterns used.

Any sequence that contained at least one base with Phred quality score less than 11 (less than 90% base call accuracy) was rejected.

**Barcode to sequence and read counts mapping.** Once the barcodes and their associated sequences were extracted and quality filtered, lookup tables with barcodes as keys, and the set of sequences (and their read counts) associated with them as values, were constructed. In many cases, a barcode was uniquely associated with a single sequence. In some cases, a barcode was associated with multiple sequences, likely as a result of sequencing errors. The three lookup tables constructed from the three gDNA read sets were merged into a single table. Pairs of lookup tables constructed from post Cas9 RNA replicate read sets were merged into single tables. Given lookup tables *T*1 and *T*2, the two are merged to form *T*3 by using the following procedure:

- Copy all barcode entries present in *T*1 but not in *T*2, into *T*3.

- Copy all barcode entries present in *T*2 but not in *T*1, into *T*3.

- For all barcode entries present in both *T*1 and *T*2, merge the set of associated sequences by taking the union of the two sets. If a sequence is present in both sets, the read counts are averaged. Copy the resulting entries into *T*3.

Let `BC_G`, `BC_PreUT`, `BC_PostUT`, `BC_PreST`, and `BC_PostST` denote tables mapping barcodes to their set of gRNA, RNA Un-Spliced (pre Cas9), RNA Un-Spliced (post Cas9), RNA Spliced (pre Cas9), and RNA Spliced (post Cas9) sequences (and their read counts) respectively.

**Sequence operation notations.** To simplify the description of dat-A and dat-B construction in later sections, we introduce the following notations and shared functions:

- Let bracket notation denote a table lookup operation, e.g. `BC_G[barcode]` refers to the set of gRNA sequences (and their read counts) associated with `barcode`.

- Let *Sim*(*Q*, *T*) denote the similarity between sequence *Q* and sequence *T*, defined as the proportion of 5-mers shared between the two sequences. More formally, modelling after the Sørensen-Dice coefficient formulation [45, 46], if *q* is the set of 5-mers in *Q*, and *t* is the set of 5-mers in *T*, then:

$$Sim(Q, T) = \frac{2|q \cap t|}{|q| + |t|} \tag{1}$$

where |*A*| denotes the cardinality of a set *A*. Thus, the output of *Sim*(*Q*, *T*) ranges between 0 (completely dissimilar) and 1.

- Let *M*(*Q*, *S*) denote the sequence in the set of sequences *S*, that is most similar to sequence *Q*:

$$M(Q, S) = \arg \max_{s \in S}[Sim(Q, s)] \tag{2}$$

As discussed in more detail in later sections, given a noisy sequence *Q* extracted from a pair of sequencing reads, we compute *M*(*Q*, lib- SA) to determine the identity of *Q*. Because there are millions of *Q* and thousands of designed sequences in lib-SA (S9 Fig and S1 Table), we chose the *Sim*(*Q*, *T*) similarity measure in lieu of exact sequence alignment for computational tractability.

**dat-A construction.** dat-A in this study refers to the set of repair outcomes observed in RNA Un-Spliced (post Cas9), their genotypes, their frequencies, and their associated gRNA (S2 Table).

For a given query target sequence $Q$ in `BC_PostUT`, a best matching designed target sequence $T = M(Q$, lib- SA target sequences) is first determined. Once the best matched $T$ is found, the designed gRNA $G$ is known by association. If $G$ is in `BC_G[barcode of Q]`, then $Q$ is associated with $G$. Otherwise, the association is not supported, and $Q$ is discarded. If $Q$ is identical to $T$, then $Q$ is also discarded as it is likely not a repair product. Otherwise, the repair genotype of $Q$ is characterized as an insertion or a deletion with a deletion size and deletion start position relative to the cutsite. This is done by simulating all possible 1-bp insertions at the cutsite, and all possible 1 to 60-bp deletions that overlapped the cutsite, on $T$. Let the set of all possible simulated genotypes be $S$, and the most similar simulated genotype to $Q$ be $s = M(Q, S)$. If $Sim(Q, s) \geq 0.3$, then $Q$'s repair genotype is characterized as $s$. Otherwise, $Q$ is discarded.

The above procedure was repeated for every target sequence in `BC_PostUT`. The resultant retained sequences were then grouped by both gRNA and repair genotype. Within each gRNA and repair genotype group, the count of the repair outcome was taken to be the sum of the read counts of all the sequences in the group.

Finally, the sequences were grouped by gRNA. Only groups that constitute at least 20 unique repair genotypes and at least 100 counts were retained. The rest were considered to have low support, and were discarded. To obtain the final frequencies of repair genotypes, the counts were normalized by the total counts within its gRNA group.

**dat-B and dat-B WT construction.**   dat-B in this study refers to the set of repair genotypes observed in RNA Spliced post Cas9 exposure, and their count of transcripts with or without Exon B (S3 Table). Similarly, dat-B WT refers to the count of transcripts with or without Exon B for each WT lib-SA target in RNA Spliced pre Cas9 exposure (S4 Table). As in the previous section, 'best matched sequence' is determined using Eq (2).

For a given transcript $Q$ in `BC_PostST` that does not contain Exon B, only the barcode information is known. To associate it with its repair genotype, its barcode is queried on `BC_PostUT`. If `BC_PostUT[barcode of Q]` is a set of one sequence, then the sequence is queried on dat-A to identify its repair genotype, and $Q$ is associated with the repair genotype. If `BC_PostUT[barcode of Q]` is a set of multiple sequences, then the sequence with the largest read count is chosen for association, and the same steps are applied. If the barcode of $Q$ is not in `BC_PostUT`, then $Q$ is discarded.

Similarly, for a transcript $Q$ in `BC_PreST` that does not contain Exon B, its barcode is queried on `BC_PreUT`, and the best matched WT target sequence is associated with it.

For a given transcript $Q$ in `BC_PostST` that contains Exon B, naturally only the Exon B portion of the repair outcome is discernible (intronic portion is spliced out). To identify its repair genotype, its barcode is queried on `BC_PostUT`. For every candidate sequence in `BC_PostUT[barcode of Q]`, their repair genotype is queried on dat-A. If only one of the candidate repair genotypes would give rise to $Q$, then $Q$ is associated with the repair genotype. Otherwise, if multiple repair genotypes could give rise to $Q$, then one of them is picked randomly, sampled according to the frequency of observing each candidate in dat-A.

For a given transcript $Q$ in `BC_PreST` that contains Exon B, its best matched WT target sequence $T$ is determined. Its barcode is then queried on `BC_PreUT`. If $T$ is in `BC_PreUT[barcode of Q]`, then $Q$ is associated with $T$. Otherwise, $Q$ is discarded.

The above procedure was repeated for every transcript in `BC_PostST` and `BC_PreST`. The retained transcripts were then grouped by repair genotype in the RNA Spliced (post Cas9) case, and by gRNA in the RNA Spliced (pre Cas9) case. Read counts were summed together within groups. Any groups with total read counts of associated transcripts of 50 or fewer were considered to have low support, and were discarded.

## Percent spliced-in

Let $T_B$ and $T_C$ be the total read counts of transcripts containing and excluding Exon B respectively. The percent spliced-in of Exon B, is then calculated as:

$$\Psi = \frac{T_B + 1}{T_B + T_C + 2} \tag{3}$$

where a pseudo-count of 1 is used. The difference between $\Psi_R$ and $\Psi_G$ (or WT $\Psi$) is that the transcripts are grouped by associated repair product and by gRNA respectively.

## inDelphi settings and evaluation

In all uses of inDelphi that produced the results presented in this study, the cell type was set to 'mESC', and genotype resolution predictions of microhomology-less deletions was turned on.

**Checking for inDelphi training data leakage.** inDelphi is trained on a dataset of 55-bp sequences, referred to as 'lib-A' in its paper [22]. The following procedure was performed to assess the similarity between inDelphi lib-A sequences and our lib-SA 61-bp sequences that make up dat-A:

1. For every lib-SA sequence in dat-A, find the most similar sequence in lib-A by performing local alignment (see below) with every sequence in lib-A, and picking the one with the highest alignment score.

2. For every best-aligned sequence pairs, compute the sequence identity (Eq (4)).

3. Save all the best-aligned sequence identities in a list.

4. Visualize the list of best-aligned sequence identities as a distribution, and characterize it using mean and median.

Local alignments were performed using the Smith-Waterman implementation provided by `pairwise2.align.localms` in Python's `biopython` version 1.74 package [47]. The same default scoring parameters used by BLAST suite's `blastn-short` program were used [48]: + 1 match, −3 mismatch, −5 gap open, and −2 gap extend.

The sequence identity ($I$) after aligning sequence $A$ with sequence $B$ ($alignment(A, B)$) is defined to be:

$$I(A, B) = \frac{\text{Number of matched positions in } alignment(A, B)}{min(length(A), length(B))} \tag{4}$$

## $\Psi_R$ prediction models and evaluation

**MaxEntScan.** MaxEntScan's [29] score3ss module was used to score a 23-nt region surrounding the acceptor splice site (20-nt of the intron, and 3-nt of the exon, at the intron-exon boundary).

**SpliceAI.** SpliceAI [23] predicts the probability a position is used as a splice acceptor for every position of a given sequence. The given sequence length is expected to be at least of length 10,000; shorter sequences can be evaluated by padding with sequences of 'N' to yield the desired length.

Each repair genotype in dat-B was preprocessed as follows before given to SpliceAI for scoring:

1. Extend the repair genotype with the intronic sequences and Exon B sequences (S1 Text).

2. Prepend and append the resultant sequence with length 5,000 sequences of 'N', to ensure the final sequence is greater than length 10,000.

After SpliceAI scoring, the maximum predicted probability over the valid positions (i.e. excluding the prepended and appended 'N' positions) of the output for each sequence was then taken to be the predicted $\Psi_R$ of each repair genotype.

**MMSplice and wMMSplice.** MMSplice [24] requires as input a reference genome FASTA file, a genome annotation file in the standard GTF format, and a variant calling format file (VCF) [49]. To represent a query repair genotype as these formats, we considered our WT lib-SA sequence as the reference sequence, and the repair genotype as the variant. That is, for each repair genotype, a FASTA file was constructed to contain our designed reference genome (S1 Text), a GTF file was constructed to contain the exonic annotations, and a VCF file was constructed to describe the 1-bp insertion or multiple base pair deletions that resulted in the repair genotype. The prediction outputs from MMSplice's 3' intronic, acceptor site, and exonic modules are log-odds values, i.e. $logit(\Psi_R)$ values, where

$$logit(x) = log(\frac{x}{1-x}) \qquad (5)$$

These values were compared against the actual observed $logit(\Psi_R)$ via Pearson's $r$.

wMMSplice is a linear model (followed by a sigmoid transform) that combines predictions from MMSplice's 3' intronic, acceptor site, and exonic modules to predict $\Psi_R$ given a repair genotype. More formally, let $\sigma$ be the sigmoid transform function:

$$\sigma(x) = \frac{1}{1+e^{-x}} \qquad (6)$$

and $logit_I$, $logit_A$, and $logit_E$ be the log-odds predictions produced by MMSplice's 3' intronic, acceptor site, and exonic modules respectively, given a sequence. The wMMSplice model for producing $\Psi_R$ predictions given the same sequence is:

$$logit(\Psi_R) \quad = \quad w_0 + w_1(logit_I) + w_2(logit_A) + w_3(logit_E) \qquad (7)$$

$$\Psi_R \quad = \quad \sigma(logit(\Psi_R)) \qquad (8)$$

The weights $w$ of wMMSplice were determined through ridge regression [30], i.e. linear least squares with L2 regularization, via the `linear_model.Ridge` module of Python's `scikit-learn` version 0.20.0 package [50]. The optimal regularization strength parameter was determined by using the `linear_model.RidgeCV` module of Python's `scikit-learn` version 0.20.0 package [50] to search for the optimal value over the space [0.001, 0.01, (0.1 to 1 at 0.05 intervals), 5, 10, 50, 100] via Leave-One-Out cross validation.

**MetaSplice.** Similar to wMMSplice, MetaSplice is a linear model (followed by a sigmoid transform) that combines predictions from SpliceAI, and MMSplice's 3' intronic, acceptor site, and exonic modules to predict $\Psi_R$ given a repair genotype (Fig 1B). More formally, let $S_p$ be the $\Psi_R$ predictions from SpliceAI, and $logit_I$, $logit_A$, and $logit_E$ be the log-odds predictions produced by MMSplice's 3' intronic, acceptor site, and exonic modules respectively, given a sequence. The MetaSplice model for producing $\Psi_R$ predictions given the same sequence is:

$$logit(\Psi_R) \quad = \quad w_0 + w_1(logit_I) + w_2(logit_A) + w_3(logit_E) + w_4(logit(S_p)) \qquad (9)$$

$$\Psi_R \quad = \quad \sigma(logit(\Psi_R)) \qquad (10)$$

To prevent undefined values of $logit(S_p)$ in cases where $S_p = 0$ or $S_p = 1$, $S_p$ values were clipped to the interval $[10^{-5}, 1 - 10^{-5}]$.

The weights $w$ of MetaSplice were determined through ridge regression [30], via the `linear_model.Ridge` module of Python's `scikit-learn` version 0.20.0 package [50]. The optimal regularization strength parameter was determined by using the `linear_model.RidgeCV` module of Python's `scikit-learn` version 0.20.0 package [50] to search for the optimal value over the space [0.001, 0.01, (0.1 to 1 at 0.05 intervals), 5, 10, 50, 100] via Leave-One-Out cross validation.

## SkipGuide model

The SkipGuide model architecture as described in this study takes as input a gRNA, a splice acceptor sequence, and associated acceptor sequence context. inDelphi is used to predict the Cas9-mediated repair outcomes and their frequencies, and MetaSplice is used to predict $\Psi_R$ from each repair genotype. SkipGuide then outputs $\Psi_G$ as an average of MetaSplice predictions weighted by inDelphi predicted frequencies of the corresponding repair genotypes (Fig 1C).

To describe this model more formally, let $R$ denote the event that the exon of interest is retained in a particular spliced transcript in a cell, and $I_R$ the indicator random variable for $R$. Thus, the value we want to predict can be interpreted as $\Psi_G = \mathbb{E}[I_R] = P(R)$, given a gRNA. If the given gRNA can causes a number of possible repair products $p_i$, then by the Law of Total Probability:

$$\Psi_G = P(R) = \sum_i P(R \mid p_i)P(p_i) \tag{11}$$

Note that inDelphi outputs $P(p_i)$ for every $p_i$, and MetaSplice takes as input a single $p_i$ and outputs $\Psi_R = P(R|p_i)$, so Eq (11) exactly describes the SkipGuide model.

## Evaluation metrics

**Cross validation.**   The independent evaluation of wMMSplice and MetaSplice, and the overall evaluation of SkipGuide was done through 10-fold cross validation, and the same 10-folds were used for both evaluations (S6 Fig). The folds were constructed by first grouping the repair outcome genotypes from dat-B by associated gRNAs, and then splitting the data into roughly equal sized folds such that no two folds contain data for the same gRNA. The `GroupKFold` implementation from Python's scikit-learn version 0.20.0 package [50] was used to perform the data splitting.

**Regression metrics.**   Let $y_i$ and $\hat{y}_i$ be the true and predicted values for sample $i$, for $i = 1 \ldots N$. The referenced mean absolute error (*MAE*), mean squared error (*MSE*), and root-mean-square error (*RMSE*) throughout this study are defined as:

$$MAE(y, \hat{y}) = \frac{1}{N} \sum_{i=1}^{N} |y_i - \hat{y}_i| \tag{12}$$

$$MSE(y, \hat{y}) = \frac{1}{N} \sum_{i=1}^{N} (y_i - \hat{y}_i)^2 \tag{13}$$

$$RMSE(y, \hat{y}) = \sqrt{MSE(y, \hat{y})} \tag{14}$$

**Binary classification metrics.** A 'positive' prediction from SkipGuide is a gRNA with predicted $\Psi_G$ less than or equal to some user defined threshold. This prediction is considered a true positive (TP) if the gRNA actually leads to an empirically observed $\Psi_G$ less than or equal to 0.5, and is a false positive (FP) otherwise.

The precision and recall scores, referenced in this study is then defined as:

$$Precision = \frac{TP}{TP + FP} \tag{15}$$

$$Recall = \frac{TP}{TP + FN} \tag{16}$$

The average precision, precision-recall curve, receiver operating characteristic (ROC) curve, and area under the ROC curve (AUC) demonstrated in this study are computed using the `average_precision_score`, `precision_recall_curve`, `roc_curve`, and `roc_auc_score` methods respectively in the `metrics` module of Python's `scikit-learn` version 0.20.0 package [50].

**Pearson's *r* and *p*-values.** All Pearson's *r* and associated *p*-values reported in this study were calculated using the `pearsonr` method of the `stats` module within Python's `scipy` version 1.1.0 package [51]. The *p*-value reported by `pearsonr` is a two-sided *p*-value under the null model of *r* = 0. A critical value for significance of $p < 0.05$ was used in this study.

## Supporting information

**S1 Fig. Genome-integrated reporter system and deep sequencing.** A schematic of the genome-integrating reporter plasmid in which a library of 61-bp human splice-acceptor intron-exon junctions is paired with Cas9 gRNA spacers. A barcode (BC) is embedded in Exon C and captured by sequencing primers (purple arrows) in genomic DNA and RNA transcripts. The deep sequencing of samples from mESCs pre and post Cas9 exposure produces the gDNA, RNA Un-Spliced, and RNA Spliced datasets, which provide barcode to gRNA association, barcode to target sequence association, and just barcode if Exon B is skipped or barcode with the Exon B portion of the target sequence otherwise, respectively. dat-A is constructed from the Post Cas9 RNA Unspliced sequences, which provide information on the genotypic outcomes and frequencies of splice acceptor repair. dat-B is derived from the Post Cas9 RNA Spliced sequences, from which Exon B skipping frequencies are elucidated. Similarly, dat-B WT is derived from Pre Cas9 RNA Spliced sequences.
(TIF)

**S2 Fig. Sequences used to train inDelphi vs. those in dat-A, and independent evaluations of inDelphi's insertion and deletion prediction performance on dat-A.** (A) The distribution of sequence identities between most similar pairs of sequences (as determined through local sequence alignment) from dat-A, and the set of sequences inDelphi was trained on (median sequence identity is 0.24). (B) The distribution of Pearson's *r* between observed and inDelphi predicted 1-bp insertion (median *r* = 0.95). (C) The distribution of Pearson's *r* between observed and inDelphi predicted deletions (median *r* = 0.75).
(TIF)

**S3 Fig. Actual vs. predicted $\Psi_R$ of 2,113 repair outcomes using MMSplice (and wMMSplice).** In all the plots, the black dashed line represents the identity line where actual equals predicted. (A) Actual $logit(\Psi_R)$ vs MMSplice intron module predicted $logit(\Psi_R)$ (*r* = 0.45). (B) Actual $logit(\Psi_R)$ vs MMSplice acceptor module predicted $logit(\Psi_R)$ (*r* = 0.20).

(C) Actual $logit(\Psi_R)$ vs MMSplice exon module predicted $logit(\Psi_R)$ ($r = 0.18$).
(TIF)

**S4 Fig. Actual vs. predicted WT $\Psi$ of 1,205 lib-SA targets using various methods.** dat-B WT originally represents 1,697 lib-SA targets, but for this analysis, only the 1,205 with at least 50 sequencing reads for estimating $\Psi$ were considered. In all the plots, the black dashed line represents the identity line where actual equals predicted. (A) Actual WT $\Psi$ vs. MaxEntScan 3' Score of the acceptor sequence ($r = 0.14$). (B) Actual vs. SpliceAI predicted WT $\Psi$ ($r = 0.6$). (C) Actual $logit(\Psi_R)$ vs MMSplice intron module predicted $logit(\Psi_R)$ ($r = 0.36$). (D) Actual $logit(\Psi_R)$ vs MMSplice acceptor module predicted $logit(\Psi_R)$ ($r = 0.24$). (E) Actual $logit(\Psi_R)$ vs MMSplice exon module predicted $logit(\Psi_R)$ ($r = 0.26$). (F) Actual vs MetaSplice predicted WT $\Psi_R$ ($r = 0.58$), where MetaSplice is tuned using the entire dat-B dataset.
(TIF)

**S5 Fig. $\Psi_G$ measured after Cas9 treatment, vs WT $\Psi$ measured before Cas9 treatment, for 735 gRNAs and their corresponding lib-SA targets.** Plot of $\Psi_G$ vs WT $\Psi$ of 735 gRNAs and their corresponding lib-SA targets. After Cas9-mediated repair, all but 23 exhibited a lower $\Psi_G$ compared to that of WT $\Psi$ (points below diagonal line). Those 23 showed only a small mean increase in PSI of 0.01. The mean WT $\Psi$ is 0.91, and shifted lower to a mean $\Psi_G$ of 0.68 after Cas-9 treatment.
(TIF)

**S6 Fig. MetaSplice (and SkipGuide) 10-fold cross validation predictions of $\Psi_R$ (and $\Psi_G$).** The same 10 folds used to evaluate MetaSplice were used to evaluate SkipGuide: the 2,113 repair outcomes in dat-B were first grouped into the 1,063 associated gRNAs, and then randomly partitioned into 10 folds. This ensures that no two folds contain repair outcomes from the same associated gRNA. Predictions for gRNAs in one fold are obtained by using the other 9 folds to fit MetaSplice's linear model weights within SkipGuide. We repeated this for all 10 folds to obtain $\Psi_R$ predictions for each of the 2,113 repair outcomes, and $\Psi_G$ predictions for each of the 1,063 gRNAs.
(TIF)

**S7 Fig. SkipGuide performance when wMMSplice or tuned SpliceAI is used instead of MetaSplice.** (A), (B), (C) The 10-fold cross validation performance of SkipGuide when wMMSplice is used instead of MetaSplice. Mean $r = 0.61$, mean $MAE = 0.17$, mean $MSE = 0.05$, mean $RMSE = 0.23$, and mean AUC = 0.81, over the 10 folds. (D), (E), (F) The performance of SkipGuide when a linear model over SpliceAI prediction (similar to that of wMMSplice and MetaSplice) is used instead of MetaSplice. Mean $r = 0.52$, mean $MAE = 0.20$, mean $MSE = 0.06$, mean $RMSE = 0.25$, and mean AUC = 0.76, over the 10 folds.
(TIF)

**S8 Fig. Perturbation studies on SkipGuide performance.** The results of a perturbation study performed on SkipGuide, where we perform the same performance evaluations depicted in Fig 4, wherein one of, or both of, the SkipGuide prediction modules are replaced with a dysfunctional predictor. (A), (B), (C) inDelphi is replaced with a predictor that always outputs uniform frequencies for all possible repair genotypes. (D), (E), (F) MetaSplice is replaced with a predictor that outputs random values between 0 and 1. (G), (H), (I) MetaSplice is replaced with a predictor that always outputs 1. Note that (I) should not be interpreted, as the precision is not actually defined at recall less than 1 (because TP = 0 and FP = 0, so precision is indeterminate). (D) Both the inDelphi perturbation that produced (A), (B), (C) and the MetaSplice perturbation that produced (D), (E), (F) were performed. Note that the precision = 1 at 0.3

threshold in (F) should not be interpreted, as precision is actually undefined at the 0.3 threshold (TP = 0 and FP = 0).
(TIF)

**S9 Fig. Read counts at each stage of sequence reads processing.** The sequence read counts retained at each stage of processing as described in the Methods.
(TIF)

**S10 Fig. wMMSplice, MetaSplice, and SkipGuide repeated 10-fold cross validation.** Depending on the data split that produces 10 folds, the evaluated performance may vary. To assess robustness, 1,000 repeats of the cross validation with random 10 fold splits were performed. This procedure would result in 10,000 fold predictions, which would provide 10,000 metric values for a given metric. The metric values shown are averages over the 10,000 metric values. SD denotes standard deviation, and precision and recall were calculated using a prediction threshold of 0.3.
(TIF)

**S1 Text. Exons and introns of lib-SA.**
(TXT)

**S2 Text. Azimuth and Basic On-Target Model (BOTM) description.**
(TXT)

**S3 Text. lib-SA cloning protocol.**
(DOCX)

**S4 Text. Regular expressions for barcode and sequence extraction.** The regular expressions used together with Python's `regex` package to extract barcodes and sequences from sequencing reads.
(TXT)

**S1 Table. The designed lib-SA target library of 1,927 intron-exon sequences and associated 4,000 gRNA sequences.** The `Designed 61-bp target site (37i-24e, AG)` column contains the designed target sequences. The `Designed gRNA (NGG orientation, 19 and 20)` column contains the gRNA sequences. The `exon_start` column contains positions relative to the Genome Reference Consortium Human Build 37 (GRCh37) [52].
(CSV)

**S2 Table. dat-A, the dataset that identifies the genotypic outcomes of splice acceptor repair.** Each row represents a repair outcome. 1,695 unique gRNAs are represented. The `gRNA ID` column consists of gRNA identifiers that correspond with the `Identifier number` column of S1 Table. The `Category`, `Genotype position`, `Inserted Bases`, and `Length` columns describe the repair genotype. The `Empirical frequency` and `Predicted frequency` columns describe the observed and inDelphi predicted frequencies respectively.
(ZIP)

**S3 Table. dat-B, the dataset that identifies skipping frequency after Cas9 treatment.** Each row represents a repair outcome, and there are 2,113 of them. The `gRNA ID` column consists of gRNA identifiers that correspond with the `Identifier number` column of S1 Table. The `Category`, `Genotype position`, `Inserted Bases`, and `Length` columns describe the repair genotype. The `Exon B Retained Count` and `Exon B Skipped`

`Count` columns describe the observed transcript counts; the $\Psi_R$ values can be calculated directly from these two columns for each row (see Methods). If the rows are aggregated by `gRNA ID`, the $\Psi_G$ values can similarly be calculated.
(CSV)

**S4 Table. dat-B WT, the dataset that identifies skipping frequency before Cas9 treatment (wild type control).** Each row represents a wild type (WT) lib-SA target splice acceptor, and there are 1,697 of them. The `gRNA ID` column consists of gRNA identifiers that correspond with the `Identifier number` column of S1 Table. The `Exon B Retained Count` and `Exon B Skipped Count` columns describe the observed transcript counts; the WT $\Psi$ values can be calculated directly from these two columns for each row (see Methods). Note that 735 gRNA identifiers are shared between this S4 and S3 Tables, so for those 735 we can compare their WT $\Psi$ and $\Psi_G$ values.
(CSV)

## Acknowledgments

We thank Benjamin R. Holmes and other members of the Gifford lab for helpful discussions and suggestions on the work.

## Author Contributions

**Conceptualization:** Wilson Louie, Max W. Shen, Christopher A. Cassa, Richard I. Sherwood, David K. Gifford.

**Data curation:** Zakir Tahiry, Sophia Zhang, Daniel Worstell, Richard I. Sherwood.

**Formal analysis:** Wilson Louie, Max W. Shen, David K. Gifford.

**Funding acquisition:** David K. Gifford.

**Investigation:** Wilson Louie, Max W. Shen, Zakir Tahiry, Sophia Zhang, Daniel Worstell, Richard I. Sherwood.

**Methodology:** Wilson Louie, Max W. Shen, Zakir Tahiry, Sophia Zhang, Daniel Worstell, Christopher A. Cassa, Richard I. Sherwood, David K. Gifford.

**Project administration:** Richard I. Sherwood, David K. Gifford.

**Resources:** Richard I. Sherwood, David K. Gifford.

**Software:** Wilson Louie, Max W. Shen.

**Supervision:** Richard I. Sherwood, David K. Gifford.

**Validation:** Wilson Louie, Max W. Shen, Zakir Tahiry, Sophia Zhang, Daniel Worstell, Christopher A. Cassa, David K. Gifford.

**Visualization:** Wilson Louie, David K. Gifford.

**Writing – original draft:** Wilson Louie, Richard I. Sherwood, David K. Gifford.

**Writing – review & editing:** Wilson Louie, Richard I. Sherwood, David K. Gifford.

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
