## [Decision Letter · Decision Letter 0]

7 Sep 2020

Dear Dr. Gifford,

Thank you very much for submitting your manuscript "Machine learning based CRISPR gRNA design for therapeutic exon skipping" for consideration at PLOS Computational Biology.

As with all papers reviewed by the journal, your manuscript was reviewed by members of the editorial board and by several independent reviewers. In light of the reviews (below this email), we would like to invite the resubmission of a significantly-revised version that takes into account the reviewers' comments. In particular, more than one reviewer raised the issue of accessibility of the source code, as well as the validation of the mESC prediction results. Please ensure that these concerns are thoroughly addressed in the revised manuscript and response letter.

We cannot make any decision about publication until we have seen the revised manuscript and your response to the reviewers' comments. Your revised manuscript is also likely to be sent to reviewers for further evaluation.

Sincerely,

Wei Li, Ph.D.

Guest Editor

PLOS Computational Biology

Jian Ma

Deputy Editor

PLOS Computational Biology

Reviewer's Responses to Questions

**Comments to the Authors:**

**Reviewer #1: **Reproducibility report has been uploaded as an attachment.

**Reviewer #2:** Deleterious Exon skipping strategy is effective o restore gene function and potentially cure some kinds of genetic disorders. Previous antisense oligo-based treatment for exon skipping requires lifespan delivery of drugs which are troublesome and expensive to the affected family. CRIPSR-Cas9 or other related Genome editing technique provide a potentially permanent solution to alleviate the symptom of a patient. However, there is no specific tool available to screen the efficient guide RNA that leads to frequent skipping of target exons holding pathogenic variants. The authors present two novel contributions to the gene therapy field with genome editing. They first develop a cell type specific genome integrated library called lib-SA as an extension of lib-A in inDelphi and preprocessing pipeline of the sequence data to quantify the Cas9 template-free repairing genotype outcome of the deleterious splicing junctions and associated exon-skipping frequency. They also provide a modular framework SkipGuide that combines the state-of-art inDelphi and MMSplice to predict the splice-in percent with a high precise performance. The library and the computational framework jointly extend the genome editing to provide an generalized therapeutic strategy to treat multiple genetic disorders.

Here are my comments for the manuscript to address the points that might not be sufficiently clear or need to be improved.

Major comments:

1. The introduction section mentioned five good examples of genetic disorder that would be hopefully cured by exon skipping. In general, how many genetic disorders so far could be potentially cured by exon skipping strategy? This information might be obtained by adding disease annotation in S1 Table, what are the diseases for the corresponding candidate pathogenic genes and sgRNA?

2. In line 246, exon A and donor site are fixed, and SkipGuide ignore the donor site contribution in the predictive pipeline, would the acceptor site effect more important than donor site in the library design principal? Or is it a simplified library which could be further extended to study the donor site sequence effect?

3. In Figure 2C, is the occurrence of AG motif in the acceptor site have any effect on the prediction? Which of the sgRNA-target pairs and associated diseases could have the best correlation between observed and prediction by SkipGuide? What the prediction correlation histogram separately for deletion and 1-bp insertion genotype since the two genotypes uses different models? In Fig. 2D, an associated scatterplot may be better to show the prediction consistency with observed frequency.

4. In Fig.3.group 10-fold cross validation may have variable performance for different sampling, multiple repeats of group 10-fold cross validation might be considered to evaluate the model robustness.

5. A clinical sgRNA would own high efficiency of exon skipping without off-target effect, what about the off-target activity from the top ranked sgRNA by SkipGuide? Is off-target activity correlated with exon-skipping frequency?

6. For the top predicted sgRNA with highest exon-skipping frequency (lowest skip-in percent) in lib-SA, what's the efficiency of exon skipping in the real transcriptomics of mESC cell line? It would be valuable to validate a few top predicted sgRNA in real exon skipping of the transcriptomics since the exon skipping would be also affected by chromatin state (H3K36me3), donor site, or other sequence context which are not considered in lib-SA.

7. In the SkipGuide pipeline, are the weights fixed for inDelphi and fine-tuned for wMMSplice?

8. For the perturbation study, what would be the precision and recall after ablating inDelphi using a similar threshold as before ablating inDelphi? What’s the performance of inDelphi model trained on another cell line other than mESC? inDelphi model with specific cell line could increase 0.07 prediction correlation, what would the performance be if the inDelphi fine-tuned on dat-A? What about the prediction correlation if the wMMSplice output is replaced with a constant 1 while only use the inDelphi model on mESC?

9. In line 382, what kind of linear model for wMMSplice, is it logistic regression model for classification? Does the model use l1 or l2 penalty for avoiding possibly colinear genotype features? How to interpret the model based on rule of the splice acceptor target sequence or sgRNA sequence composition?

10. The most advantage of CRIPSR-Cas9 over antisense oligo treatment is the permanent treatment effect. In lib-SA, Cas9 transfects the mESC with Blasticidin selection to maximize Cas9 activity. How would the Cas9 be delivered to maximize its activity in the long term for a clinical setting? This might be discussed in the discussion section to provide a practical insight of using lib-SA and SkipGuide.

Minor comments:

• Full name of MMSplice should be written as modular modeling of splicing, and the full name for wMMSplice should be also listed in the main text.

• In Fig 1A and S1 Fig, it would be better to annotate the dat-A and dat-B sequences on the workflow schematics.

• In Fig.1 and S1 Fig, there is no intron between exon B and C in lib-SA plasmid, would exon C be skipped as well in RNA splice sequence?

• In benchmark dat-B, only MMSplice and MaxEntScan are evaluated, how about other method mentioned in the introduction, e.g. SPANR?

• In line 110, the main text description is not consistent with the Fig. 2B, Fig. 2B show the observed frequency of inserted based at -4 bp, however, the text describes the consistency of the predicted frequencies and observed frequency.

• In line 162, after binarizing the observed skip-in values, what would be ROC curve and the Area Under the Curve (AUC) look like?

• In later library design section from Methods, which version of HGMD, Ensembl and Pfam were used?

• How are the sgRNA on-target threshold of BOTM and Azimuth decided for designing lib-SA?

• Methods on preprocessing lib-SA to get repaired genotype outcome and splice-in percent are the key part to generate benchmark dataset for SkipGuide, this part might be improved to be more clear: 1. it might be more accessible to the reader if separating the shared function for matching query sequence to wild type sequence and calculating similarity into a subsection before data-A/B construction with pseudo-code; 2. In equation (1), the denominator should be the intersection between q and t to evaluate the similarity; 3. In line 292, BC_PreST is not described; 4. in line 300, is the WT target sequence T from (Pre Cas9 BC_PreUT), and what's the advantage of using 5-mers matching strategy instead of using smith waterman alignment? 5. In equation 2), are the percent spliced-in computed after the read counts normalization within sgRNA? 6. In 394 line, the VCF described both the indel and insertion type of the WT target.

• What's of the remaining proportion of reads count after each step of preprocessing, from regex operation, barcode with multiple sequences, and dat-A/B construction? Such a table could be provided to give a global view of lib-SA quality.

• Original fastq and supplementary datasets should be deposited to GEO or EBI or public lab websites.

**Reviewer #3:** In this study, Louie et al. developed a novel pipeline, SkipGuide, for predicting CRISPR gRNA editing effects for exon skipping. The pipeline consists of their own inDelphi model for CRISPR editing outcome prediction and wMMSplice for exon-inclusion level prediction. Using a reporter library of 791 splice sequences and 1063 gRNAs in mESC, the authors demonstrated the high precision of SkipGuide for finding gRNAs with effectively skipped target exons.

This work addresses an essential biological problem in bridging CRISPR-Cas9 therapeutics and the splicing machinery, even though the methodological novelty is low. I would like to see it published in PLOS computational biology, pending some concerns as below.

1. As a practical proof-of-concept, it will greatly strengthen the paper’s claim if at least some endogenous exon-skipping targets’ predictions in mESC are validated.

2. The splicing PSI prediction is subpar especially for PSI_R. Since the goal is to predict the absolute level of PSI, using Pearson’s correlation to evaluate a predictor’s performance is insufficient. The authors should also report metrics like mean squared/absolute errors and try a few other predictors such as SpliceAI (Jaganathan et al., 2019).

3. Figure 3 indicates a strong upward bias for predicted PSI_R. The authors did not address this bias nor discuss its influence on downstream analysis (which clearly persists in Figure 4). This can potentially undermine the practical utility of this work.

4. Figure 4 shows the high precision of predicted PSI_G, and also reveals the high False Negative rate for SkipGuide. Do these false negative predictions share any sequence and/or CRISPR editing characteristics in common?

5. The authors should demonstrate that the training data used to train inDelphi (Shen et al. 2018) was not overlapped nor homologous to dataset dat-A. In particular, since inDelphi was set to run using mESC mode in this work, it’s important to show there is no training data leakage from a machine learning perspective.

6. It’s unclear from Figure 2B how inDelphi’s 1bp insertion model was evaluated. The panel only shows the empirical distribution for 1bp insertions.

7. Code for this work (claimed to be on GitHub) is not publicly available for evaluation.

**Have all data underlying the figures and results presented in the manuscript been provided?**

Reviewer #1: None

Reviewer #2: **No: **The supplementary processed dataset only available to reviewer, the software is not available in public site yet. Original fastq and supplementary datasets should be deposited to GEO or EBI or public lab websites.

Reviewer #3: Yes

PLOS authors have the option to publish the peer review history of their article (what does this mean?). If published, this will include your full peer review and any attached files.

Reviewer #1: **Yes: **Anand K. Rampadarath

Reviewer #2: **Yes: **Qian Qin

Reviewer #3: No
---

## [Decision Letter · Decision Letter 1]

3 Dec 2020

Dear Dr. Gifford,

We are pleased to inform you that your manuscript 'Machine learning based CRISPR gRNA design for therapeutic exon skipping' has been provisionally accepted for publication in PLOS Computational Biology.

Best regards,

Wei Li, Ph.D.

Guest Editor

PLOS Computational Biology

Jian Ma

Deputy Editor

PLOS Computational Biology

Reviewer's Responses to Questions

**Comments to the Authors:**

Reviewer #1: Reproducibility report has been uploaded as an attachment.

Reviewer #2: This revised version of the manuscript completely addressed all my previous questions in a nice way, and improved the main text with additional supported figures, tables and references accordingly. The methods section are largely improved, it is now clear about how to reproduce the data analysis with the release of Github code and sequencing downloading links. This work will be definitely very helpful for the clinicians to provide new avenue of designing exon-skipping-based CRISPR-Cas9 therapy to treat known and future new-identified rare disease.

However, there are two last minor points to be corrected: 1) 163th line S2 Fig label should be replaced with S3 Fig label. 2) If I understand Dice coefficient correctly, equation 1) between line 403 and 404 should use intersection symbol in the numerator. 3) one additional comma in line 418.

Reviewer #3: The authors have addressed all my comments and concerns.

**Have all data underlying the figures and results presented in the manuscript been provided?**

Reviewer #1: None

Reviewer #2: Yes

Reviewer #3: Yes

PLOS authors have the option to publish the peer review history of their article (what does this mean?). If published, this will include your full peer review and any attached files.

Reviewer #1: **Yes: **Anand K. Rampadarath

Reviewer #2: **Yes: **Qian Qin

Reviewer #3: No

---

## [Editor Report · Acceptance letter]

31 Dec 2020

PCOMPBIOL-D-20-01290R1 

Machine learning based CRISPR gRNA design for therapeutic exon skipping

Dear Dr Gifford,

I am pleased to inform you that your manuscript has been formally accepted for publication in PLOS Computational Biology. Your manuscript is now with our production department and you will be notified of the publication date in due course.

With kind regards,

Jutka Oroszlan
